# "Authenticity" as a Pathway to Sustainable Cultural Tourism? The Cases of Gotland and Rapa Nui

**Marije Eileen Poort** [1,*] , **Ulrika Persson-Fischier** [1] , **Helene Martinsson-Wallin** [2] , **Evelina Elf Donaldson** [1] and **Mareike Schaub** [1]

[1] Department of Civil and Industrial Engineering, Uppsala University, 75237 Uppsala, Sweden; ulrika.persson-fischier@angstrom.uu.se (U.P.-F.); Evelina.Elfdonaldson.5662@student.uu.se (E.E.D.); Mareikekerstin.Schaub.0507@student.uu.se (M.S.)

[2] Department of Archaeology and Ancient History, Uppsala University, 62167 Visby, Sweden; helene.martinsson-wallin@arkeologi.uu.se

\* Correspondence: marije.e.poort@angstrom.uu.se

**Abstract:** In this paper, two World Heritage island destinations are compared, Gotland in the Baltic and Rapa Nui in the Pacific. Both islands deal with a growing number of tourists, which asks for rethinking of tourism development. As cultural heritage plays a major role in tourism for both destinations, it is especially important to look into sustainable cultural tourism development from the perspective of authenticity. The paper is based on a variety of materials and takes an innovative approach to methods through a student–staff collaboration. The data show that social acceptance of tourism by local communities on both islands could be achieved through the creation of authentic experiences. Furthermore, data show that it is rather potent to engage both locals and tourists in modernized authentic experiences. However, this can only be done if locals are empowered and are genuinely allowed to steer the development of tourism.

**Keywords:** authenticity; sustainable tourism; liquid modernity; Gotland; Rapa Nui

## 1. Introduction

Sustainability studies have often focused on islands, as islands are "good to think" with as a metaphor for the whole Earth as an island in the universe, and because as demarcated entities their fate can provide us with valuable insights into sustainable and unsustainable practices. Rapa Nui (Easter Island) in the Pacific has often been used to illustrate what we can learn about unsustainability and collapse of environmental and social systems [1–3]. Rapa Nui has been regarded as an example of an unstable society that collapsed, but as all sustainability issues, this is a complex story. When Europeans first came to Rapa Nui, they met a treeless landscape, but in contrast to the generalized image of a collapsed society the first ethno-historic account by the Dutch commander Roggeveen [4] paints a picture of an island that was extensively cultivated and fertile and that the people prospered. Geological and archaeological investigations have later shown that the island was covered with an extensive palm tree forest along with a handful of endemic trees and bushes prior to human arrival [5]. The extensive clearing of land was carried out for agricultural purposes and to make room for the multitude of now well-known ceremonial sites that currently are great tourist attractors. The cutting down of the forest paved way to the collapse of to the original ecosystem, however new inventive agricultural methods actually made the society resilient. The meeting with the Europeans was in fact a major factor to the disaster and destruction. We now know that the newcomers fueled social unrest among the Rapanui, which ended up in the destruction of agricultural fields and monuments [6–8]. Due to the colonial impacts along with the changed ecosystem and biodiversity loss, the population diminished. The unsustainable dimensions are indeed facts, however from the perspective of the Pacific Island people, the one-sided story that

the society collapsed due to their carelessness of managing their environment could be viewed as an additional form of colonialism of an indigenous people [7,8]. What happened in Rapa Nui depends on a set of complex factors that are lessons to be learned of human impact on ecosystems and the devastating impacts of colonialism. Despite colonization, forced relocation and unsuited large-scale husbandry, people have continued to make a living on Rapa Nui. Today, the main livelihood on Rapa Nui is tourism, as tourists are attracted to the unique history and cultural heritage of places like Rapa Nui.

The island Gotland is different from Rapa Nui in many ways; situated in the global north rather than global south, and hence a very different socio-economic context. It is larger in size, and with some more livelihood activities than only tourism. However, there are also many similarities between both islands; tourism is the main economic activity, and cultural heritage is the main attraction for tourists. Furthermore, both islands house UNESCO cultural heritage sites. Even if Gotland is much larger than Rapa Nui, the ratio between inhabitants and tourists is the same as on Rapa Nui. Just as Rapa Nui, Gotland has a "colonial" history of a kind in relation to mainland Sweden, to which it is socio-economically disadvantaged. Both islands are vulnerable in similar ways, depending on fragile environments and cultural resources that are needed for, but also threatened by economic development, in the shape of tourism and over-tourism.

Even after what has been discussed in terms of the collapse, populations are able to sustain. This indicates that we need to ask more detailed and complex questions that include environmental, socio-cultural and economic dimensions of sustainable development, to explore how we can arrive at societies that keep within planetary and social boundaries for better future lives. Studies on sustainable development have in this way progressed in holistic understandings on the impact of human behavior and activities on the environment. However, as Suntikul argues, "an equivalent concern for the impact on cultural inheritance remains a relatively underarticulated perspective" [9] (p. 2103), especially relevant for tourism destinations whose main attractions are the tangible and intangible cultural heritage. This also means that this kind of sustainability study cannot solely rely on natural scientific quantitative research, but needs to involve the perspectives of people, as it is people that are involved in the development and possible transformation of culture and tourism.

As both islands deal with a rapid growth of (cultural) tourism, the focus of this article is on tourism and cultural heritage, comparing the two island destinations. This has resulted in contemporary situations on the islands, which also pose sustainability challenges to them. Tourism gives economic possibilities to marginalized places all around the world, not least islands, in situations where very few other economic possibilities exist. In situations like Gotland and Rapa Nui, where the cultural heritage is the main tourist attraction, the very basis for the islands' marginalization, their cultural vulnerability also becomes a strength in how to overcome its vulnerability. On the other hand, with too much tourism there is a strain on the heritage sites, which run the risk of being worn down. In such situations, tourism runs the risk of destroying what attracted tourism to the destinations in the first place. Moreover, too much tourism can be a social strain for the local population, who might feel their home place becomes overcrowded, and that their local culture is not respected. Furthermore, there is the risk that too much tourism demands too many natural resources, like fresh water, and that local ecosystems and infrastructure cannot deal with many people visiting temporarily. Sewage and trash are common problems of this kind. Islands are especially vulnerable in this way, since it is not so easy to import more resources from elsewhere.

This asks for rethinking on how to develop tourism sustainably, especially with a focus on the cultural aspect. Figueroa and Rotarou [10], Tiberghien et al. [11] and Suntikul [9] argue that involvement of the local community and the offer of authentic experiences to cultural visitors are of main importance for sustainable cultural tourism development. It offers tourists a sense of a "genuine" experience and it can create greater acceptance of tourism within the local community, which is the foundation for sustainable tourism.

Suntikul [9] stresses the importance of cultural sustainability from a "liquid modernity" perspective, in which the modern and the traditional are not static but fluid and constantly evolving when enacted. This means that identities are always evolving, as "cultural values within which individuals and communities are embedded constantly shift" (p. 2104). Traditionality and modernity are thus interwoven. Cultural sustainability is characterized as "locally defined and culturally embedded relations and meanings" [12] (p. 328), and deals with "the recovery and protection of cultural identities" [13] (p. 31). In tourism, culture is often commodified. This can threaten the cultural sustainability [14]. Participation by the community in tourism development could therefore play an important role in making sure that cultural heritage expressions do not become diluted and superficial and that a sustainable cultural tourism development is supported [9]. A "liquid modernity" form of tourism thus opens up for co-creation of authentic cultural experiences, in which tourists and locals alike participate on more equal terms and gain equally.

By comparing the two island heritage destinations trough a qualitative research model, we aim to get insights into what extent authenticity in this form is relevant for the people living on these islands. We investigate how the local community is and can be involved in tourism development in order to create a foundation for sustainable tourism development.

## 2. Gotland and Rapa Nui

Rapa Nui (Easter Island) situated in the East Pacific Ocean and Gotland in the Baltic Sea are major tourist destinations. They represent small island communities and their major tourist attractions are enigmatic, well-known archaeological sites and monuments. On Gotland, the medieval city if Visby became a World Heritage site in 1995 and in the same year Rapa Nui's National Park with its numerous giant stone statues and ceremonial sites was nominated as World Heritage.

Gotland is a 3184 km$^2$ large island situated in the Baltic Sea proper. It belongs to Sweden and currently has around 60,000 inhabitants. Each year a little over a million tourists visit the island. It has a colonial history and the city of Visby was part of the Hanseatic League in the Middle Ages with a strong influence of a German population. In the 15th century Gotland came under the Danish rule, which changed in the 17th century when the island came under the Swedish rule. For a long time, it was a poor and neglected part of Sweden. However, due to the exotic nature and many historical monuments, it slowly became a tourist destination in the 19th–20th centuries. Today it is one of the most visited tourist destinations in Sweden and therefore, it has developed social-economic wise. Mainland–island and rural–urban tensions have existed for a long time. This finds its origin in the colonial history and the Visby battle of 1361, where the rural community (Gutnish country yeomen) fought the Danish troops and were left to fend for themselves by the citizens of Visby [15]. This tension still exists now, as the focus of tourism mainly is on Visby and its surrounding areas [16]. Gotland does not have an indigenous people but when local communities and authenticity in relationship to cultural heritage are discussed in the Gotlandic context, it is entangled with several factors. Firstly, it is linked to the question, who is considered to be Gotlandic, which has to do with if you are born and raised on the island. Secondly, it is linked to speaking the specific Gotlandic dialect. Thirdly, it has to do with you being considered a genuine Gotlandic person (Gute). For this to be true, your ancestors have to have lived on the island for five generations. Furthermore, it can be debated what specific cultural heritage is on Gotland; the cultural heritage that is highlighted for the tourists is mainly the ringed wall of Visby.

Rapa Nui is a small speck of land (c. 164 km$^2$) in the East Pacific Ocean. It was settled around 1100 years ago by Polynesians, has a violent colonial history and was annexed by Chile in 1888 [5,17]. In the beginning of the 20th century only a little over hundred Indigenous people (Rapanui) remained on the Island. Today Rapa Nui has around 7500 inhabitants of which c. 1500 consider themselves indigenous Rapanui. Tourism started in the 1960s, subsequent to the world-famous explorer Thor Heyerdahl's archaeological expedition to the island in 1955–56 [18]. Wealthy tourists have mainly been attracted to

Rapa Nui due to its giant stone statues (moai) and the many histories of the past population. In 2018, administration of the World Heritage national park was handed over from the Chilean State organization of National Parks (CONAF) to the Indigenous organization Ma'u Henua. There are around 100,000–150,000 tourists that visit Rapa Nui each year (as of 2019), of which the majority are Chileans. There has been and still are tensions between the Rapanui and the Chilean State and some Rapanui demand independence [6]. Figueroa and Rotarou [10] state there is a perceived loss of Rapanui identity, and that locals feel threatened by "Chileanization" as mainland Chileans who come to work in the tourism industry become permanent residents. On the other hand, Delsing [7] states that the historical oppression has strengthened the Rapanui people in defining their own identity, take pride in their culture and fight for its preservation. One expression of Rapa Nui culture and its preservation is dancing. Fieldwork reveals that dancing is expressed in at least two different ways. On the one hand, Rapanui dance to other Rapa Nui (and other Polynesian people), in local festivals, et cetera. This is to express, experience and strengthen their own culture and cultural identity. The meaning of these dances is internal to their culture and not immediately available to external viewers. On the other hand, Rapanui dance to tourists. In the evenings, there are performances at restaurants to which tourists can buy tickets. Interviews with dancers reveal that the content of these dances are the same. The dances to tourists are not adjusted or different than dances to other Rapanui. This probably means that is not evident that tourists—who lack the cultural background—understand the dances in the same way as Rapanui people do.

## 3. Methods

In order to be able to compare such diverse destinations as Gotland and Rapa Nui, we bring together a variety of material in an innovative approach, built upon a students–researchers collaboration. Our data build on theses work of students in the master program Sustainable Destination Development from the spring of 2020, with a focus on heritage tourism on Gotland and Rapa Nui [19,20], from students interviews on Gotland in 2020 (Gansauer and Poort), from in-depth fieldworks both on Gotland, extensively between 2017–2020 (Persson-Fischier) and Rapa Nui in November 2019 (Martinsson-Wallin, Persson-Fischier and Poort) and on the long-term engagement in tourism and heritage development on both islands of Helene Martinsson-Wallin on Rapa Nui and Ulrika Persson-Fischier on Gotland. This paper is thus the result of a collective work over time. During the fieldwork on Rapa Nui in November 2019, the authors did observations and conducted 25 semi-structured interviews with various tourism stakeholders, both public and private. The interviews took 1–1.5 h and there was a local present to translate to and from Spanish. The interviewees were chosen based on the recommendations and local knowledge of our local translators regarding tourism on Rapa Nui. To make the data comparable, the same kind of actors and stakeholders were identified and interviewed through similar semi-structured interviews of 1–1.5 h on Gotland by a student of the master program Sustainable Destination Development (see Table 1). Because of the Covid pandemic, most interviews took place online and fewer interviews were conducted, as it was harder to contact and involve the actors. For this reason, Table 1 only shows the corresponding actors of both destinations and it is the interviews with these actors that are analyzed in this research. The semi-structured interviews contained questions regarding collaboration between tourism actors on both destinations, the importance of sustainability in the actors' activities and outlooks on development of the destinations. The raw data were analyzed by an inductive, clustering methodology through qualitative content analysis by students Demuro, Gansauer and Van der Zee, under the guidance of Poort.

**Table 1.** List of corresponding tourism actors.

| Respondents Easter Island | Respondents Gotland |
| --- | --- |
| Sernatur | Tourism information |
| Tour Agency | Destination Gotland |
| Diving Center | Bike rental |
| (Cruise) tour agency | Cruise services |
| Tourism Chamber | Gotland Convention Bureau |
| Museum | Museum |
| Archaeologist/guide | Archaeologist/guide |
| Guides | Gotland guides association |
| SECPLAC | Region Gotland |

In addition to the fieldwork, our data derive from two student theses. The thesis of Elf Donaldson [19] is based on a content analysis of Trip Advisor reviews of six cultural tourism businesses on Rapa Nui: three tour companies and three dance groups. These data represent how visitors to Rapa Nui perceive and value authenticity when engaging in cultural heritage tourism. Elf Donaldson selected reviews that spoke about authenticity, relationships to the culture, the need for information, the local heritage, the interactions with guides and storytelling that were posted within the last five years. This led to around 30 relevant reviews per businesses. The thesis of Schaub [20] is based on semi-structured interviews with five tourism and heritage development experts on Gotland. The interviews focused on the experts' views of heritage tourism, which heritage sites are visited and challenges and potentials of this tourism, within the framework of sustainable development.

## 4. Sustainable Tourism and Authenticity

### 4.1. Sustainable Cultural Tourism

Sustainability can be interpreted in various ways. Strong sustainability primarily focuses on maintaining the functional aspect of ecosystems, whereas weak sustainability focuses on reducing negative environmental impact while using natural capital as a resource [21]. There are also various definitions of sustainable tourism, of which the one of the UNWTO is often used: "Tourism that takes full account of its current and future economic, social and environmental impacts, addressing the needs of visitors, the industry, the environment and host communities" [19].

Saarinen [22] identifies three traditions of sustainable tourism, the first with a "tourism first" perspective, mainly focusing on the regeneration of the industry, i.e., economic sustainability. This perspective does not take into account the fact that economic development of tourism often tends to create environmental sustainability challenges and does not take the locals and their needs into consideration. The second tradition focuses on environmental sustainability, for example, in terms of carrying capacity, the number of tourists a destination can carry before environmental problems arise [22]. This view though has difficulties in accounting for how tourism and other activities have intertwined consequences, where it is difficult or impossible to discern what actually is the consequence of tourism and that as tourism comes with other economic activities it is impossible to find the ecological tipping points only for tourism activities.

In the third tradition, the needs of the community in which tourism takes place are taken as a point of departure. Crucial to this tradition is to include the local community and have democratic decision making in place. In tourism development, there needs to be a constant re-evaluation of the rationale for engaging in tourism, vis-à-vis other possible activities. Tourism is one of the many activities in which local communities can engage [22]. This third tradition is especially relevant for heritage tourism, as the involvement of local communities is crucial for sustainable cultural tourism development [9,10]. The concept of authenticity is important, as cultural tourists seek "genuine" experiences and authenticity can create a higher degree of acceptance of tourism by locals. "This is particularly true in the case of islands, which are characterized by fragile ecosystems and limited size, since the

increased interaction between tourists and residents can reveal more easily any negative impacts caused by tourism development" [10] (p. 247). Furthermore, several studies argue that the local communities, or in other words the hosts of the destination, are at the heart of tourists' experiences [23]. The locals' quality of life on all levels of sustainability thus needs to improve with the development of (cultural) tourism.

### 4.2. Identity, Authenticity and Tourism

In recent years, tourism saw a rapid growth on both Rapa Nui and Gotland. It is therefore of importance to be aware of the negative and positive impacts of tourism, within the contexts of the islands. Tourism can be more than only an economic driver and can give destinations a certain leverage in case of disputes or tensions and give pride to historically devalued communities [24]. Moreover, it can make local communities (more) aware of their identity and the value that lies in their cultural heritage [25]. This in turn can be beneficial for tourism development, as locals are the ambassadors of heritage and can deliver an authentic experience which cultural tourists are looking for [9–11]. It is important to keep in mind that tourism often entails the commodification of the local culture and traditions. Cohen [14] (p. 372) states that this could lead to the exploitation and destruction of these local cultures and traditions.

Authenticity is relevant, as culture and history are a major attractor of tourists on both Rapa Nui and Gotland alike. Tourist come to see the archaeological sites, dances and festivals, where visitors can effectively seek out the authentic, "a modern value defined by notions of pristine, natural or untouched culture" [24] (p. 98). Although there is no universally accepted definition of the concept, a distinction can be made between objective and existential authenticity. Objective authenticity refers to "an inherent feature of objects" and the manner in which tourists attribute authenticity to objects, whereas existential authenticity concerns any "personal connection" the tourist has with the destination cultivated through participation, mostly in activities that are presumably part of the locals' everyday lives [26] (p. 249). The existential tourist is one who spiritually abandons modernity, moves furthest away from the beaten track and tries to get as close as possible to the "Other" [14] (p. 377). Both island locations that we used as case studies house tangible World Heritage sites, the medieval ringed wall on Gotland and the National Park with the ceremonial sites (ahu) and their large stone statues (moai), on Rapa Nui. These are major tourist attractors and it is often believed that the conservation and management of tangible heritage sites is "in itself fulfilling a sustainable development objective" [27] (p. 46). However, ever since the ICOMOS Nara charter on authenticity from 1994, there have been discussions on that cultural heritage, authenticity and sustainability are more complex matters. Labadi [27] suggests that a holistic view is necessary which includes the consideration of local participation in, and benefit from, heritage protection. The Budapest declaration from 2002 was the first UNESCO official document which addressed sustainable development and heritage. In the Operational Guidelines for the Implementation of the Convention from 2005, it is stated that World Heritage properties "may support a variety of ongoing and proposed uses that are ecologically and culturally sustainable" [28] (p. 33) The last ten years there have been serious efforts to integrate the sustainability concept in the nomination process and management of World Heritage sites but the implementation has halted. For a detailed account and discussion on these matters see Labadi [27] (pp. 45–60) and UNESCO Operational guidelines 2019 [28]. The development of Intangible Cultural Heritage has contributed to a more complex discussion on authenticity, cultural heritage and sustainability. Because intangible heritage is constantly recreated, the concept of "authenticity" applied to World Heritage properties cannot be used for ICH [20]. However, in a tourist situation the intangible heritage as for example dance and language can be perceived by the visitors as more or less authentic. Another example of the complexity of these concepts is that tangible cultural heritage, in the case of the Rapanui monuments used in tourism, has been rebuilt, reused and is a materialized ideology, and the ideology and use have changed over time [29]. Another distinction that can be made is the staged

and the backstage authenticity. "Tourist settings can be viewed as a continuum, with the foremost region being the one that is for show and the backmost region the one that is considered more authentic and motivates touristic consciousness providing a "chance to glimpse the real". The "backstage" region where hosts' genuine cultural heritage is maintained and cultural integrity and identity is kept is the intimate and authentic part of the tourism destination that is sought by some visitors" [11] (p. 289). They thus argue that most tourists mainly see a "performative authenticity", the staged version of a local culture.

For a "true" authentic experience, there needs to be a genuine host–guest relationship. An intimate connection between the transmitter of the culture and the tourists creates within the tourist a deeper understanding of the heritage and "a sense of closeness and a story about a shared experience" [11] (p. 290). Existential and backstage authenticity are thus, more than objective authenticity, of major importance for the satisfaction of the cultural visitor [11,26]. Furthermore, and as said before, it fosters locals' acceptance of tourism and strengthens local identity and ownership of local heritage [9–11]. This is where authenticity from a liquid modernity perspective comes in, as it is a way to think about authenticity that allows for both tradition and historical roots and change and improvement of living conditions. It is constantly evolving as enacted, which means that both locals and visitors become co-creators of authentic experiences [9].

## 5. Results and Analysis

In this section we describe and analyze the findings of the multiple data collections mentioned in the methods section. First, we outline the data collected by Elf Donaldson [19], then we describe the data found by Schaub [20], and lastly, we list the results of interviews with actors on Rapa Nui and Gotland (see also Table 1). These results are categorized in themes derived in an inductive manner from the data.

### 5.1. Results of Trip Advisor Reviews of Rapa Nui Visitors

The tourists' reviews of cultural businesses in Rapa Nui show that cultural visitors mainly look at the who when it comes to "measuring" authenticity. Being indigenous Rapanui as a guide or performer proves to be an indisputable sign of authenticity. This gives the guides and performers the authority to provide accurate information and enables them to embody and represent their culture.

When it comes to the performance venues in particular, the tourists seem to appreciate enthusiasm and dedication to the songs and dances. It is as if a dedication to island traditions and art forms themselves is the determining factor in ensuring an "authentic" manifestation of local culture, as if they are not inclined to sacrifice their cultural pride and integrity for the sake of commodification in the financial interests of commercial tourism. In the case of the tour companies, the visitors seem to value the relationship between themselves and the guides. By sharing personal and familial stories, the guides establish a temporary friendship with the visitor.

Both of the above-mentioned features are especially beneficial for the culture-oriented tourist who seeks an immersive experience to discover and understand local heritage. The reviews of all six tourism businesses show that a perception of empowerment through cultural manifestations exists, and that visitors view the indigenous Rapanui as having agency over the way their culture is portrayed within the tourism industry. Furthermore, the cultural-oriented visitor seems to appreciate their own contribution to the safeguarding of heritage and empowerment of the local population through their presence as a tourist even more than the authentic experience itself [19].

### 5.2. Results of Interviews Tourism and Heritage Experts Gotland

The interviews with experts on Gotland show that the local community is seen as of major importance for tourism, as the locals are the ambassadors of the island. The experts say that host-guest relationships are maybe even more important than Visby as a cultural heritage town itself. Not only the interactions in tourism settings are relevant, but also the

interactions outside of these settings, at places of local importance for example. Residents of Gotland are in this way contributing in an essential way to the cultural experience of the visitor, and existential authenticity thus plays a major role in cultural tourism on the island.

Most tourists coming to Gotland visit the island for a combination of purposes. The heritage is for a small group of people the only reason to visit. Most tourists, however, like to combine the heritage with nature and the sun and the beach. It is the domestic tourist who mainly comes for the nature and bathing purposes, international visitors seem to be more interested in the cultural heritage.

On Gotland there are to be found around 60 to 70 small museums, which are run by local heritage associations or non-profit organizations. The experts identify these local actors as crucial for the tourism experience, as it is these associations and organizations that make the rural heritage available for tourists. Most of them want to actively take part in tourism, and tell their story to visitors. The associations are thus important in shaping the locals' identity and in providing spaces for meaningful host-guest relationships. However, their resources are limited as it is mostly volunteers who do the work. This makes them rather unprofessional and economically often not viable. An improved management of these organizations could be helpful, however taking the management too much out of the hands of the locals could mean a loss of identity and ownership, which can have implications for the acceptance of tourism [20].

*5.3. Results In-Depth Interviews with Tourism Stakeholders on Both Islands*

5.3.1. Identity

The interviews at both island destinations show some similarities and differences in their outlooks on and application of authenticity in tourism. A very important difference, also mentioned earlier in this article, is the existence of an indigenous community and living indigenous culture on Rapa Nui. These kinds of traditional values of course influence how much value is placed in cultural identity and authenticity, especially taking into account the recent colonial history and the marginalization of the indigenous culture this entails.

> " . . . heritage for us is much broader . . . we Rapanui are part of the living heritage, we have our language, our traditions, our culture . . . and we do not have the same vision as in other places about heritage . . . we are still alive, we still carve our moai, we maintain our traditions . . . what we want is to continue developing and for me the future would be that tourists who come to Rapa Nui and also take part, not only come to take the photo, but to learn and teach us . . . " —Representative Ma'u Henua

On Gotland such a(n) (living) indigenous culture does not exist, and the cultural heritage is less intertwined with the current identity. This means that, although important for both islands, tourism on Rapa Nui serves not only as an economic driver but could preserve the indigenous culture. Nevertheless, and as emphasized by Tiberghien et al. [11], on Gotland tourism can also be in more general terms a transmitter of local cultural heritage and the current living culture. A local guide on Gotland describes:

> *You know, Gotland is different. But I think it's also part of the charm. You know, we have the older generation, we have the mentality of, that the island. Everything goes. It's alright. Which is a good thing. It's difficult, because it's good!*

Being on an island and being isolated from the rest of Sweden has thus created an identity of being different, of an "everything goes" mentality. This "other" culture could also be transmitted through tourism on Gotland.

5.3.2. Objective Authenticity

Both Rapa Nui and Gotland mainly attract tourists through their cultural heritage as "objects", which translates to artifacts such as the moai on Rapa Nui and the Visby ringed wall on Gotland. The objective authenticity entails the perceived originality of these artifacts. As the originality of the cultural heritage on both islands is proven by archaeological research, and underlined by the World Heritage status, there does not seem

to be much discussion around the objective authenticity. This is in line with what the experts say, as the host–guest relationship is more important than the World Heritage town Visby itself.

The importance of originality probably correlates with the type of tourist (and the kind of experiences this tourist is looking for), shows an example on Rapa Nui. A diving school has placed a moai made of concrete in the ocean which with divers can take a photo. Divers are told or already know it is not an original moai, but still enjoy visiting it and taking pictures with it. Here the originality does not seem to be important, as diving tourists main aim for visiting is not experiencing the cultural heritage but diving in special places all around the world. This is what the owner of the diving shop said about it:

*"The real one is not found yet. We know where maybe it is, but we don't can reach the yeah [ . . . ] this one. It's made from concrete [it is from] the neighbors. So, the history is for the memory of the grandfather of the owner of the other dive center. So, they wanted to do something beautiful to him and to the tourist, you know. [ . . . ] So many people come to see it. [ . . . ] And we tell it's from concrete, it's okay and they want a photo. So, it's good, it's good [ . . . ] It's good for tourism too, because something different is a llamativo [remarkable]"*

This is a good example of authenticity from the perspective of liquid modernity. The moai is not real or original, still the locals and tourists have co-created an authentic experience of it. An example of this on Gotland can be found in the Medieval week. This week is based on the real history of Visby, however the experiences are staged and enacted to relive the medieval times and meet the present needs of visitors. Another example is the cartoon garden that shows famous cartoon figures which are not genuine cultural heritage for Gotland. Visitors are aware of this, but they still go there as it is attractive to families with children.

### 5.3.3. Cultural and Social Sustainability

A challenge regarding the cultural "objects" is the preservation. On Rapa Nui the archaeological sites in the National Park are under the administration of the Rapanui organization Ma'u Henua, which means the sites in the National Park are only accessible with a valid ticket and at certain times. However, this brings tensions as the local Rapanui want to be able to visit their ancestors whenever they want. Because of the opening hours, the local people feel restricted in accessing their own culture. Another challenge regarding the preservation of the cultural "objects" is the herding of free-roaming cattle. As the farmers on the island do not use fences, the animals can walk on the archaeological sites. This can damage the cultural "objects", and could therefore become a problem for attracting visitors. Being able to communicate objective authenticity and focus on developing cultural sustainable tourism can thus lead to socially unsustainable situations where locals are or may be restricted in their practices.

On Gotland a similar situation exists regarding the cathedral in Visby, as the place tries to serve two purposes. Locals would like to exercise their religion in quiet and peace, while visitors want to visit the famous cathedral and use—one of the few—public toilets located in the cathedral. From a sustainable cultural tourism perspective, it is logical to show the cathedral, as it is an important part of the historical artifacts in Visby. However, this again brings challenges regarding the wider social sustainable situation. A high degree of acceptance of the local community could counter this kind of tensions, in which authenticity beyond the objects can play a major role.

### 5.3.4. Existential Authenticity

Existential authenticity seems to be a concept of more importance on both islands, as this is about the personal connection a tourist can have with the destination.

*"In the first thing archeology impresses him [the tourist], it is like when one goes in Egypt and sees the pyramid . . . thus another pyramid, another moai, another moai and then a*

*point arrives although his interest is rather towards living culture, getting to know people . . . yes -dancing, shows . . . -more than dancing, living with people"* —Representative tourism office Rapa Nui

When talking about existential authenticity, the discussion on staged and backstage authenticity becomes relevant, as this is about genuine host–visitor connections. On Rapa Nui the local culture is not only communicated via guides on tours; an important aspect is the dance groups, as underlined by the data collection of Elf Donaldson. These groups show the indigenous dances from the island and the Polynesian area. As stated by a local tourism business owner these shows are the same for tourists and locals.

*"[ . . . ] then she said, then she noted 'O, though I must mention we don't see it as-as if we are doing our performance for tourists'. That's what she said. 'You know, we are doing it for-we are showing our culture, but it is not necessarily for tourists'. And-uhm they-they are the oldest group of the Island Kari Kari. And they especially have a very uhm I'd like-they are the most like authentic. You know, most true Rapanui dance, you know. And-uh they are-they don't. She said, you know, she said it is exactly the same, you know, if they do it for tourists or locals. But, all of the groups are like that though. Here on the Island. They have even more focus, no but still uh all the groups are like that. The dancers, the people perform here. They are always gonna be the same if it's for tourists or for locals, you know. The music and dance is so of this Island, you know. So, there is no need to-to fake it, really."*

He further exemplifies that in his businesses he always tries to give tourists as much of a local experience as possible. Then he also states that by giving these local and authentic experiences to tourists, value can be created which goes beyond just the tourist perception.

*"Yeah I think, as I mentioned before, you know, the same day. I think more than putting value, showing things means that you value to it, you know. [ . . . ] Going to the elders uuh that is putting value to the elders. You know, and people say. The elder is gonna say 'wow they're coming here? From all over the world to see me? I'm 80 years old, I can barely walk. But they see me as important, the see me as special.' You know, and then everyone else will start to follow that example, also seeing that person as special, being a tourist or not".*

This tourism business owner is not the only one offering local experiences, other actors try to offer or have a wish for development of local experiences. This is something a representative of Ma'u Henua emphasizes as well:

*"I would perhaps like there to be the opportunity that the tourist will cook with us [ . . . ] I mean, they get to know our culture and not only the heritage that is around . . . we, we are heritage, we have recognition [ . . . ] yes, like what they call ethno-tourism . . . that's what they call the approach, when the tourist . . . that is when tourism focuses on knowing the place where you live".*

On Rapa Nui the focus thus seems to lie on the backstage authenticity rather than the staged. Furthermore, this underlines the outlook on tourism as a driver for cultural and personal pride.

Comparing this to tourism offers in Gotland, we see a similarity with the train trip from Dalhem. Tourist can see the local environment and pass an elderly home, where many habitants wave as the train passes by. In addition, on Gotland there are tours taking the visitors to various local places to see how the products are made and to taste or buy these local products. A local guide talks about a cheese maker, for example:

*"But the cheese from Starvae, it's the best ever. And people buy it. He has a little shop there. And people go in after they had been presented. They get a taste plate, and he explains what they make and how. And then, they have a moment to buy".*

This is a clear example of how a farmer tries to expand his activities by sharing his practices and his locally produced cheese with visitors. Other respondents bring up the

fact that the rural communities are realizing the benefits tourism can have for them. A local guide says:

> "[b]ecause people are getting into marriages and they're coming here. And the, what it does is that it opens up chances. Out in the countryside, they're working so hard, because. To find ideas for trips. I mean, we go out on farms, to show sheep, and the wool and all of that".

Creating a genuine and local host-guest interaction is thus a strategy of many actors on both islands. Many are creating existential authentic experiences, because they see a win-win situation. This is in line with what Schaub writes, as there are many active small heritage associations run by volunteers. These groups do not only take on the lion share of taking care of the cultural objects, many of them seek to have more tourism in order to share their stories and heritage.

### 5.3.5. Commodification

As mentioned by Cohen [14], a threat of tourism can be the commodification of identity and heritage. Even though there are some examples to be found of existential and backstage authenticity, another heard answer from tourism stakeholders on Rapa Nui is that tourism businesses and the industry lack a real connection with the cultural heritage, since their main aim is [to earn] money. A representative of the National Heritage Board says the following:

> "They [tourism businesses] consider heritage as a product [ . . . ] there is not any concrete program trying to reconnect Rapanui elders with their own heritage and to allow the appropriation of the community on their own heritage. So, that's why they [tourism businesses] just want to be full of tourists without any other concern. Now the only concern for them, it's their own income".

While a tourism business owner mentions the following:

> "At this moment the park is managed as a business, which in some sense is fine for it to be like that but it is not just a business. It is necessary to have an awareness for the group that manages the park, to have an awareness that this is a cultural heritage not just business".

This is of course very much entangled with the existence of a local indigenous culture that is still alive today and it connects to the earlier discussed challenge of tourism development and management and social sustainability. Another example of commodification caused by tourism is the fact that more and more land on Rapa Nui is built. A representative of Ma'u Henua explains about how tourism has affected the Rapanui identity:

> "That is, we have lost, I said it, we have lost the value of the land, we have left ours aside . . . that is, how do I explain it? We have perhaps put aside the value of the land, the value of space, the space where we live because everything is already built, everything is . . . all spaces are built and there is no longer a place to build in the town and you have to go to other places further away . . . ".

The traditional connection of Rapanui with the land is thus affected by the tourism development.

From the outlook of theory, wherein identity can be preserved and developed through tourism and wherein cultural tourists seem to seek for genuine, authentic experiences [9–11,26], one could say that the tourism stakeholders should focus more and better on the link between heritage and tourism activities and experiences. However, it also looks as if various tourism actors are pointing fingers to each other when it comes to commodification of culture. As Elf Donaldson [19] found out, most visitors actually experience the activities and tourism services as authentic. Perhaps the difference here is that it is the locals' culture that is commodified for tourism, so the tourism actors are afraid that others do not protect the living indigenous culture enough. This could have to do with liquid modernity as well; influenced by modernity, local people need to constantly adapt as traditions are constantly

changing. So, authenticity then becomes a never-ending debate and the questions on "what is our identity" and "how should it be communicated" always stays relevant. Nevertheless, it seems that creating existential authenticity and backstage experiences are positive for preserving the local culture and adding value to the businesses.

## 6. Discussion

In this paper the focus lied on two UNESCO world heritage island destinations, Rapa Nui in the Pacific and Gotland in the Baltic. Both destinations deal with a rapid growth of tourism, which asks for rethinking how to develop tourism sustainably. Furthermore, although different, both islands have a history of colonization and the local communities have historically been devalued. Tourism can bring this value back and help rural and indigenous communities in re-shaping their identity, and for this reason these local communities need to be involved in tourism development [9].

Since both islands rely much on cultural tourism, it is important to look into the concept of authenticity. From the literature we learn that especially existential authenticity and so-called backstage experiences enhance the satisfaction of the cultural visitor by creating a genuine host-guest relationship. Simultaneously, it can positively influence the acceptance of local communities for tourism development and strengthen identity [9,10].

From the various materials we have analyzed in this study, we see a similar pattern. Existential authenticity seems to be something visitors are looking for and something tourism stakeholders on both islands are striving for. On Rapa Nui this is done by guides that tell personal stories and dance groups that see their shows as merely a communication of their living culture than a show for tourists. On Gotland local heritage associations maintain the rural heritage and try to be involved with tourism and locals (in the rural areas) develop more and more products and services to be offered to tourists. There seems to be a wish for more host-guest interactions and to share authentic local experiences.

However, there are some challenges too. Various actors on Rapa Nui see the commodification of their indigenous culture as a threat, the focus seems to mostly lie on the business rather than the cultural heritage. This shows the relevance of the concept liquid modernity and the need for a constant debate on what local culture is and means. On Gotland professionalism and financial support misses when it comes to the heritage in the rural areas, most work is done by volunteers.

Therefore, and following the argumentation from the literature, it could be beneficial for both Rapa Nui and Gotland to implement the cultural aspect of sustainability clearly in their tourism strategy. However, this needs to go beyond the artifacts and cultural "objects". Aiming for both cultural and social sustainability means to involve the (rural or indigenous) community and to put effort in developing rural/local tourism experiences, with the focus on genuine host-guest interactions. This leads to existential and backstage authenticity, something cultural tourists are often looking for [11] and which enhances their satisfaction [26].

Developing experiences that focus on engagement and create a personal connection could not only benefit the tourist but also the local community and the tourism businesses. As we see on Rapa Nui, the local indigenous culture is communicated in an original way, to both locals and tourists. This is a way to preserve the culture and to create cultural pride. For Rapanui this could mean a leverage towards the Chilean government, in pursuing rights and acknowledgement and money to invest in preservation and development projects. Gotland does not have a dispute like this, however it can be translated towards the ongoing tension between the island and the mainland and between the rural and urban community. Sustainable cultural tourism could especially be beneficial for the rural community, as they could get more leverage towards the region and receive investments to develop their local area. Regarding the tourism businesses, a better tourist experience adds value to the company and could thus lead to more and/or better paying customers.

Social acceptance of tourism can be achieved through authenticity. By having positive and genuine host-guest interactions and a high level of backstage authenticity, the accep-

tance of the local community regarding tourism can enhance. This goes hand in hand with the before mentioned tensions between tourism development and management and the exercising of local culture. There needs to be attention from authorities and businesses for the value of cultural heritage, which can be communicated through authentic tourism experiences. Furthermore, our data show that it is rather potent to engage both locals and tourists in modernized authentic experiences. This can potentially result in problems, as powerful actors could just simply introduce completely new things for tourists that do not have any historical or cultural links with the destination. However, if locals are empowered and are genuinely let to steer the development, it could be a productive co-creation, which benefits both locals and tourists. All in all, involvement of local communities and offering existential and backstage experiences can result in a win-win situation, where the local communities get pride, revive their identity, share local knowledge on history and heritage, while simultaneously creating a better tourism experience which adds value to businesses and the destination as a whole. This leads to a higher degree of social and cultural sustainability in the destination.

From our study a few recommendations for practical destination development can be drawn. As both tourists and hosts value and search for existential authenticity, and since neither tourists nor host groups want commodification of culture, this constitutes a potential point of departure for destination developers and DMOs. When attempting to develop new tourist products and services, a first step can be to involve the local community, and seek to understand how they understand their culture, from a "liquid modernity" and existential authenticity perspective. Based on this new products and services can be developed, that both tourists and hosts will aspire and feel comfortable with. This is also one way to safeguard socially and culturally sustainable development of tourist destinations.

**Author Contributions:** Conceptualization, M.E.P., U.P.-F., H.M.-W., E.E.D. and M.S.; methodology, U.P.-F.; resources, H.M.-W.; validation, H.M.-W.; investigation: M.E.P., U.P.-F. and H.M.-W.; data curation, M.E.P.; writing—original draft preparation, M.E.P., U.P.-F., H.M.-W., E.E.D., M.S.; writing—review and editing, M.E.P.; project administration, U.P.-F.; funding acquisition, H.M.-W. All authors have read and agreed to the published version of the manuscript.

**Funding:** STINT funded the travel and accommodation expenses to and in Chile/Rapa Nui.

**Institutional Review Board Statement:** Not applicable.

**Informed Consent Statement:** Informed consent was obtained from all subjects involved in the study.

**Data Availability Statement:** Not applicable.

**Acknowledgments:** Mattarena Tuki, Marcus Edensky, Olivia Gustavsson, Monica Frisk, Karin Winsnes, Mats Jansson.

**Conflicts of Interest:** The authors declare no conflict of interest.

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
