# Peer review of "“Authenticity” as a Pathway to Sustainable Cultural Tourism? The Cases of Gotland and Rapa Nui"

_sustainability, doi:10.3390/su13116302_

Round 1
Reviewer 1 Report
The article presents an accurate perspective of sustainable tourism, my congratulations. A correct approach and methodology. Some aspects that could be improved are:
- The number of interviews carried out, number of workshop participants ... justification for selecting the profiles of the informants should be specified in the methodology.
- It is important to describe how these interviews have been processed. The table provided by the informants would be interesting to provide more data, for example, number of interviews for each group, sex, date of completion ...
- The results obtained from the workshop are not clearly reflected.
Reviewer 2 Report
Dear authors,
Thank you very much for the hard work on the manuscript. I read with interest your paper. The student-staff collaboration and the focus on local communities needs and aspirations are highly appreciated. However, the article tackles too many concepts with little elaboration which might confuse the reader and hinder the manuscript's understandability. Please find below a couple of suggestions aimed at improving the flow and readability:
-In your introduction you describe how the two islands were impacted in the past but you don't elaborate on why they need to be sustainable today and what are the negative externalities of tourism. The only refrence to that is
"Both islands are vulnerable in similar ways, depending on fragile environments and cultural resources that are needed for, but also threatened by economic development, in the shape of tourism and over-
tourism."
It would be useful to elaborate on what are the negative impacts. This will help drawing some conclusions on how to transform the threats into opportunities. It might be useful indeed to use a SWOT analysis to help the reader understand the issues at stake.
-the discussion about authenticity is very interesting but since you are dealing with two UNESCO WHS, it is advisable to enrich your literature review of how the concept of authenticity evolved in the UNESCO and International conventions and charters and what scholars from ICCROM and ICOMOS debated and continue to debate today, for example see the work of Jukka Jokilehto, Gamini Wijesuriya, Herb Stovel, etc...
-On cultural heritage and sustainability, please have a look also on the work of Sophia Labadi where the four pillars of sustainability (cultural, social, environmental and economic) are taken into consideration. See also, Larsen, P. and Logan, W. (eds.) (2018), World Heritage and Sustainable Development: New Directions in World Heritage Management. London: Routledge; Labadi, S. and Logan, W. (eds.) (2015), Urban Heritage, Development and Sustainability. London: Routledge
-Please elaborate more on the two case-study, for example, the indigenous dances from the island and the Polynesian area are very representative and different from one to another, a pragraph describing what they narrate and how is relevant.
-it is clear that a lot of time and effort was dedicated to this research but the description of data collection and the discussion need to be enhanced. For example, it is not clear what type of interview took place (structured, semi-structured, open, etc..), What were the questions, what was the main research hypothesis? what were the main findings, etc...
Thank you and good luck,
Reviewer 3 Report
The paper is interesting as a case study and an approach to sustainability and authenticity. However, methodologically it needs to be more detailed and include some tables with the axes of the interviews and the form of analysis of the discourses/answers. Likewise, interviews or more indirect assessments should be included regarding the opinion of the local community less directly involved in tourism (for example, education or health sector, another basic economic services or the elder population, etc). Consequently, it is very biased and needs a review from perception studies and population profiles consulted.
Concerning the references and background, must be reviewed UNESCO sources about Tourism management in WH sites, and concerning management plans and community involvement in WH in the Operational guidelines of WH convention.
ICOMOS Document of Nara must be cited and consult to debate the auntenticy https://www.icomos.org/charters/nara-e.pdf
Concerning analysis of another studies about Tripadvisor and this topic they can consult for example : Corpas, N and Castillo, A. PASOS; 2019 Vol.: 17 Núm.: 1 Págs.: 39-52 DOI: https://doi.org/10.25145/j.pasos.2019.17.003
This is enough to understand the paper needs an important revision to be published. Besides, the conclusions are generals. They need to be more proactive in them, introducing more specific proposals to improve the subject of study.
Round 2
Reviewer 3 Report
This version is improved, but they can work a little more conclusions and proposal